# CD8^+^ T Cells Mediate Lethal Lung Pathology in the Absence of PD-L1 and Type I Interferon Signalling following LCMV Infection

**DOI:** 10.3390/v16030390

**Published:** 2024-03-01

**Authors:** Alanna G. Spiteri, Tamara Suprunenko, Erin Cutts, Andrew Suen, Thomas M. Ashhurst, Barney Viengkhou, Nicholas J. C. King, Markus J. Hofer

**Affiliations:** 1Viral Immunopathology Laboratory, Infection, Immunity and Inflammation Research Theme, School of Medical Sciences, Faculty of Medicine and Health, The University of Sydney, Sydney, NSW 2050, Australia; 2Charles Perkins Centre, The University of Sydney, Sydney, NSW 2050, Australia; 3School of Life and Environmental Sciences, The University of Sydney, Sydney, NSW 2050, Australia; 4Sydney Cytometry, The University of Sydney and Centenary Institute, Sydney, NSW 2050, Australia; 5The University of Sydney Institute for Infectious Diseases, The University of Sydney, Sydney, NSW 2050, Australia; 6The University of Sydney Nano Institute, The University of Sydney, Sydney, NSW 2050, Australia

**Keywords:** lymphocytic choriomeningitis virus (LCMV), T cell exhaustion, lung pathology, type 1 interferon, IFNAR1, PD-L1

## Abstract

CD8^+^ T cells are critical to the adaptive immune response against viral pathogens. However, overwhelming antigen exposure can result in their exhaustion, characterised by reduced effector function, failure to clear virus, and the upregulation of inhibitory receptors, including programmed cell death 1 (PD-1). However, exhausted T cell responses can be “re-invigorated” by inhibiting PD-1 or the primary ligand of PD-1: PD-L1. Further, the absence of the type I interferon receptor IFNAR1 also results in T cell exhaustion and virus persistence in lymphocytic choriomeningitis virus Armstrong (LCMV-Arm)-infected mice. In this study, utilizing single- and double-knockout mice, we aimed to determine whether ablation of PD-1 could restore T cell functionality in the absence of IFNAR1 signalling in LCMV-Arm-infected mice. Surprisingly, this did not re-invigorate the T cell response and instead, it converted chronic LCMV-Arm infection into a lethal disease characterized by severe lung inflammation with an infiltration of neutrophils and T cells. Depletion of CD8^+^ T cells, but not neutrophils, rescued mice from lethal disease, demonstrating that IFNAR1 is required to prevent T cell exhaustion and virus persistence in LCMV-Arm infection, and in the absence of IFNAR1, PD-L1 is required for survival. This reveals an important interplay between IFNAR1 and PD-L1 with implications for therapeutics targeting these pathways.

## 1. Introduction

Infection with noncytopathic lymphocytic choriomeningitis virus (LCMV) elicits several divergent host responses, depending on the dose and variant of LCMV, the inoculation route, and the mouse strain used [1,2,3]. The critical balance between the rate of virus replication and the development of cell-mediated immunity in naïve individuals determines whether the LCMV is cleared or persists [3]. Invasive strains of LCMV, including Docile and clone 13 (CL-13), produce higher viral loads that result in T cell exhaustion, virus persistence, and chronic infection [3,4]. Conversely, infection with less-invasive variants such as Armstrong (Arm), WE, and Aggressive, leads to an acute disease that is efficiently cleared by virus-specific CD8^+^ T cells within two weeks [3,5,6]. These distinct models of LCMV have been critical to our current understanding of anti-viral immunity, interferon (IFN) signalling, T cell exhaustion and chronic infections. Indeed, LCMV was the first model used to uncover the role of PD-1 in T cell exhaustion [4]. This discovery has since been extended to other chronic infections and has contributed to the development of innovative therapeutics for malignant tumours.

In the acute phase of LCMV-Arm infection, antigen-specific CD8^+^ T cells undergo massive clonal expansion. These generate effector cells that produce cytokines such as IFN-γ and TNF, as well as cytolytic enzymes, granzyme B, and perforin, all of which contribute to virus clearance [5,7]. The differentiation of naïve CD8^+^ T cells to functional effector CD8^+^ T cells requires three signals, including: (1) T cell-receptor (TCR) engagement, (2) co-stimulation, and (3) cytokines such as interleukin 12 and type I IFNs (IFN-I) [8].

Type I IFNs, constituting 14 IFN-α subtypes and 1 IFN-β subtype in mice, are a family of anti-viral cytokines produced in response to virus infection [7,9,10,11]. They promote innate immune defence and the development of functional CD8^+^ T cell responses [7,9]. We have previously shown that in LCMV-Arm-infected mice, the absence of IFN-I signalling in mice deficient in the IFN-I receptor, IFNAR1, results in the reduced accumulation of virus-specific T cells with a limited ability to produce effector cytokines [6], highlighting the importance of IFN-I to a functional CD8^+^ T cell response in LCMV.

In contrast to LCMV-Arm, infection with LCMV-CL13 results in virus persistence due to the exhaustion of CD8^+^ T cells [4]. This phenotype is characterised by the expression of multiple inhibitory molecules such as PD-1, TIM3, LAG3, CTLA4, and TIGIT and a reduced capacity to proliferate and produce effector cytokines (e.g. TNF and IFN-γ) or granzyme B upon stimulation [4,6,12,13]. Of the inhibitory molecules, PD-1 and its receptor PD-L1 are particularly important in the context of LCMV. PD-1 is an inhibitory receptor that suppresses T cell activation by interfering with co-stimulation and/or TCR signalling [14]. Importantly, blocking the PD-1/PD-L1 pathway with a monoclonal antibody is sufficient to “re-invigorate” exhausted CD8^+^ T cells in chronically LCMV-CL13-infected mice, resulting in reduced viral load [4,5,15]. Due to its central role in mediating T cell exhaustion, the PD-1/PD-L1 pathway has been exploited by a number of cancer therapeutics [16,17,18,19].

As outlined above, we have previously shown that IFNAR1 signalling is critical to the generation of functional CD8^+^ T cells in LCMV-Arm infection. Therefore, in the current study, we aimed to investigate the role of PD-L1 signalling in the generation of exhausted T cells and virus persistence in LCMV-infected IFNAR1 knockout (KO) mice. Utilizing IFNAR KO, PD-L1 KO, and double-KO animals (PD-L1 and IFNAR KO, DKO), we reveal that IFNAR1 and not PD-L1 is required for a functional CD8^+^ T cell response, while PD-L1 was required to prevent lethal lung pathology in IFNAR1 KO mice. This study reveals a significant interplay between IFNAR1 and PD-L1 signalling, highlighting its implications for therapeutics that target these pathways in the context of chronic infections and cancer.

## 2. Materials and Methods

### 2.1. Mice

All mice were of a C57Bl/6 genetic background and were bred under specific pathogen-free conditions at the animal facility of The University of Sydney. Ethics approval for all experiments was obtained from the animal ethics committee at The University of Sydney (approval numbers 1056/2016 and 1738/2020). Animal experiments were performed in compliance with the NSW Animal Research Act (and its associated regulations) and the 2004 NHMRC Australian code of practice for the care and use of animals for scientific purposes. All mice used were 8–25 weeks of age and age-/gender-matched as best as possible. IFNAR1 KO mice have a targeted gene disruption of the type I interferon (a/b) receptor 1 gene *Ifnar1* and were kindly provided by Michael Frese, Australian National University, ACT. PD-L1 KO mice have a targeted gene disruption of *Cd274*— which encodes PD-L1, (alternative names: B7-H1, CD274) and were kindly provided by Lieping Chen, Johns Hopkins Medical Institute, Yale School of Medicine. IFNAR1 x PD-L1 DKO mice were generated by crossbreeding IFNAR1 KO and PD-L1 KO mice.

### 2.2. LCMV Infection

Mice were intraperitoneally infected with 1 × 10^3^ or 2 × 10^5^ plaque forming unit (PFU) of LCMV-Armstrong 53b (LCMV-Arm). This route of infection was used to allow for a better comparison to our previous study [6]. All experiments analysing clinical disease, weight, histology, and gene expression changes were conducted with a low-dose infection. The high-dose infection was used for flow cytometry experiments as the higher dose results in more virus-specific CD8^+^ T cells, a focus of this study. No differences in clinical disease were observed between both infection doses. The LCMV-Arm stock was originally obtained from a triple-plaque-purified clone that was subsequently passaged twice in BHK cells in M. Oldstone’s laboratory [20] and provided by I. L. Campbell [21]. LCMV stocks were diluted to the required PFU immediately before infections in a virus diluent consisting of PBS + 2% foetal bovine serum. Following infection, mice were observed daily, and weight and clinical scores were recorded. Clinical scores were determined by adding up individual scores based on an animal ethics committee-approved scoring scheme (rough fur = 1, hunched posture = 2, reduced activity = 2, tremor = 3, and seizures = 4). Mice were euthanised by isoflurane inhalation on the days indicated unless they met humane endpoint criteria for euthanasia, i.e., 20% weight loss or clinical score of 4 or more.

### 2.3. Monoclonal Antibody Neutralisation of CD8, Ly6G and PD-L1

To deplete neutrophils, mice were treated with anti-GR1 (RB6-8C5, BioXcell, #BE0075; days 3, 4, 5, 6, 7, 8 p.i.) and anti-Ly6G (1A8, BioXcell, Lebanon, NH, USA, #BE0075-1; days 3, 5, 7 p.i.) intraperitoneally at a dose of 200 μg prepared in 200 μL of sterile PBS. Isotype control mice were depleted with one or both isotype control monoclonal antibody(ies) (mAbs) (LTF-2 and 2A3, respectively, BioXcell, #BE0090 and #BE0089) on the same days and in the same amounts. CD8^+^ T cells were depleted with anti-CD8 isolated in-house from 2.43 hybridoma cell supernatant at days 0, 2, 4, 6, and 8 p.i. at a dose of 200 μL of sterile PBS. A neutralising antibody against PD-L1 was prepared in-house from MIH5 hybridoma cell supernatant. The concentration of the neutralising antibody was determined using the Rat IgG total ELISA Ready-SET-Go!^®^ kit (Affymetrix eBioscience, San Diego, CA, USA). For PD-L1 neutralisation experiments, mice were injected with 200 μL (~270 μg) of neutralising antibodies three times weekly beginning on day 15 or 24 post-infection (p.i.) for a period of 2 weeks, as indicated in the figure legend. For flow cytometry experiments, control mice were injected with 200 μg of Rat IgG2a isotype control, anti-trinitrophenol clone 2A3 (BioXCell, #BE0089).

### 2.4. RNA Extraction and Real-Time Quantitative Polymerase Chain Reaction

For RNA extraction, lung, liver, and brain tissue was dissociated with TRI Reagent (Sigma Aldrich, Cleveland, OH, USA) using a tissue homogenizer (IKA, Nordrhein-Westfalen, Germany). The RevertAid RT Reverse Transcription Kit and random hexamer primers (ThermoFisher Scientific, Boston, MA, USA) were used to generate cDNA, and the SensiFAST SYBR Lo-ROX (Meridian Bioscience, Cincinnati, OH, USA) was used to conduct qPCR on the LightCycle^®^ 480 Instrument II (Roche, Basel, Switzerland). Primers are listed in Appendix A. Gene expression values were normalized to 18S or Rpl13a.

### 2.5. RNase Protection Assay

The RNase protection assays (RPAs) were performed as described previously [21,22]. The probe set used to quantify LCMV RNA was prepared by combining EcoRI-linearized pGEM-4Z plasmids containing a 450-base fragment of the LCMV-NP gene (NP_694852.1) and a 78 bp fragment of the ribosomal protein L32-4a (L32). Furthermore, [α-32P]-UTP radiolabelled antisense probes were generated by in vitro transcription. RNA samples (15 μg) were hybridised in an excess probe. Hybridised samples were treated with single-strand specific RNases (1 μg/mL RNase A, 10 mU/mL RNase T1), and remaining RNA fragments were separated by electrophoresis on a 5% polyacrylamide gel. The gel was transferred onto Whatman paper and bands were visualised by autoradiography on CL Xposure film (ThermoFisher Scientific) or on Phosphor Screens (Amersham Biosciences, Little Chalfont, UK) and visualised on a Typhoon FLA9000, (GE Life Science, Marlborough, MA, USA). For quantification, densitometry analysis of each band was performed using ImageJ software v1.48 [23]. The densitometric value for each transcript signal was normalised against the corresponding L32 RNA loading control.

### 2.6. Histology of Liver and Lung Tissue

Tissues for histological analysis were removed and placed in ice-cold PBS-buffered 4% paraformaldehyde (pH 7.4, Sigma-Aldrich, St. Louis, MO, USA) for 24 h at 4 °C and washed out with 70% ethanol. Following paraffin embedding, 5 μm thick tissue sections on Superfrost^®^ slides (ThermoFisher Scientific) were prepared and stained with haematoxylin-eosin (H&E). Stained sections were examined under a DM4000B bright field microscope (Leica, Wetzlar, Germany). Bright field images were acquired using a SPOT Flex 15.2 64 Mp Shifting Pixel camera and SPOT Advanced 4.5 software (Diagnostic Instruments, Sterling Heights, MI, USA).

### 2.7. Quantification of Plasma Cytokines

Plasma cytokines were measured using the LEGENDplex™ Mouse Anti-Virus Response Panel (Biolegend, San Diego, CA, USA, #740622) according to the manufacturer’s instructions. Samples were acquired on an LSR-Fortessa X-20 flow cytometer (BD Biosciences, Franklin Lakes, NJ, USA) and analysed using the software provided. The LEGENDplex™ assay was repeated once, giving comparable results.

### 2.8. Flow Cytometry

Blood was collected from anaesthetised mice by cardiac puncture using a 1 mL syringe and 26G x 1/2” needles coated with 1000 U/mL heparin (Sigma-Aldrich). The spleen and lung were collected from anaesthetised mice after perfusion with ice-cold PBS. Spleens were gently mashed through a 70 μm nylon mesh sieve using a syringe plunger. Lungs were processed into single-cell suspensions in RPMI and DNase I (80 U, Sigma-Aldrich, USA) and collagenase D (2 mg/mL, Sigma-Aldrich, USA) using the gentleMACS dissociator (Miltenyi Biotec, Bergisch Gladbach, DE) with default programs “m_lung_01” and “m_lung_02”. Homogenates were placed in a water bath for 30 min at 37 °C between dissociation programs and subsequently passed through a 70 µm mesh cell strainer. RBC lysis buffer (Invitrogen, Carlsbad, CA, USA) was used to lyse erythrocytes in a single-cell suspension of blood, spleen, and lung. After tissue processing, live cells were counted with trypan blue (0.4%) on a haemocytometer. Single-cell suspensions were incubated with purified anti-CD16/32 (Biolegend, USA) and UV-excitable LIVE/DEAD Blue (UVLD) (Invitrogen) or the Zombie UV Fixable Viability kit (Biolegend) and subsequently stained with a cocktail of fluorescently labelled antibodies (See Appendix A). Cells were washed two times and fixed in a fixation buffer (Biolegend).

For intracellular cytokine staining (antibodies listed in Appendix A), splenocytes were incubated in the presence of 1 μg/mL LCMV GP33-41 (KAVYNFATM) or 1 μg/mL NP396-404 (FQPQNGQFI) peptide (JPT Peptide Technologies, Berlin, Germany) and 10 μg/mL brefeldin A (Sigma-Aldrich) for 4 h at 37 °C and 5% CO_2_. Following this, cells were blocked, stained for surface antigens, and fixed as described above. Cells were then permeabilised using Perm/Wash buffer (#554714, BD Biosciences) for 15 min at room temperature (RT) and stained for intracellular antigens in Perm/Wash buffer for 30 min at 4 °C. Cells were washed twice in Perm/Wash buffer before being resuspended in FACS buffer for analysis.

Splenocytes were measured using the FACSDiva programme on an LSR-II fluorescence-activated cell sorter (FACS) (Becton Dickinson, San Jose, CA, USA), while lung and blood samples were measured on the five-laser Aurora Spectral Cytometer (Cytek Biosciences, Fremont, CA, USA). The acquired data were analysed in FlowJo (v10.8, BD Biosciences). Quality control gating including time, single cells, non-debris/cells and Live/Dead staining was applied to exclude debris, doublets, and dead cells. Cell numbers were quantified using cell proportions exported from FlowJo and total live cell counts.

### 2.9. tSNE Analysis

The FCS files were compensated and gated in FlowJo prior to exporting channel values (CSVs). T-distributed stochastic neighbour embedding (tSNE) was applied to CSV files in RStudio (1.1.453 or 1.4.1717) using Spectre [24] (package publicly available: https://github.com/ImmuneDynamics/Spectre using default settings, accessed on 2 February 2023), employing default settings, i.e., perplexity = 30, theta = 0.5, and iterations = 1000.

### 2.10. Statistical Analysis

Non-parametric statistical tests were applied to the data using GraphPad Prism 8.4.3 (GraphPad Software, La Jolla, CA, USA). A comparison of the two groups was conducted using the Mann–Whitney test, and three or more groups were compared using a Kruskal–Wallis test with a Dunn’s multiple comparison test. When two independent variables and three or more groups were being compared, a two-way ANOVA and a Šídák’s or Tukey’s multiple comparisons test was used. A Log-rank (Mantel–Cox) test was used to determine the statistical significance of survival data. Error bars are shown as standard error of the mean (SEM).

## 3. Results

### 3.1. Absence of IFNAR1 and PD-L1 Signalling Results in Lethal Disease following LCMV Infection

LCMV-Arm infection in wild-type mice is self-limiting [3,4,5,6,25]. By contrast, as previously shown by us, the absence of type I interferon signalling in IFNAR1 KO mice results in a chronic infection characterised by CD8^+^ T cell exhaustion and the expression of PD-1, virus persistence, and transient disease [6]. To elucidate the role of PD-1 and IFNAR1 signalling in LCMV infection, we generated DKO mice by ablating the expression of *Cd274*—which codes for PD-L1, the primary ligand of PD-1—and *Ifnar1*. Intriguingly, following intraperitoneal LCMV infection, DKO mice died by day 10 p.i., while wild-type (WT), IFNAR1 KO, and PD-L1 KO mice survived (Figure 1a). Clinical scores of DKO mice increased from day 7 p.i., with mice showing hunched posture and a significant reduction in body weight (Figure 1b,c). This enhanced morbidity and mortality in DKO mice cannot be explained by an increase in viral load, as there was no significant difference in LCMV glycoprotein (LCMV-GP) or nucleoprotein (LCMV-NP) RNA in the lung, liver, or central nervous system (CNS), compared to IFNAR1 KO mice (Figure 1d,e), although we did observe a non-significant increase in viral RNA in the livers of DKO mice. Additionally, lethality cannot be explained by the enhanced systemic production of a range of pertinent cytokines in DKO mice (Appendix A), as there were no significant increases in these in the blood, compared to IFNAR1 KO mice, which develop only mild disease and survive LCMV infection. Thus, in the absence of IFNAR1, PD-L1 is required for survival during LCMV infection.

### 3.2. IFNAR1-Deficient Mice Fail to Accumulate LCMV-Specific CD8^+^ T Cells Independently of PD-L1

Given that CD8^+^ T cells in the spleens of IFNAR1 KO mice exhibit an exhausted phenotype, which contributes to enhanced virus persistence in LCMV infection [6], we examined these cells in DKO mice to determine if PD-L1 has a role in T cell exhaustion in this model. Firstly, we examined splenocytes in DKO mice for virus-specific CD8^+^ T cells using dextramers specific for the LCMV-GP and LCMV-NP (Figure 2). Interestingly, relative to WT mice, both IFNAR1 KO and DKO mice showed a substantial reduction in both the number of CD8^+^ T cells and virus-specific CD8^+^ T cells (Figure 2a–c). However, this effect was clearly not dependent on PD-L1 since PD-L1 KO mice showed no difference in the number of CD8^+^ T cells relative to WT mice (Figure 2a–c). This indicates that PD-L1 does not contribute to the failed accumulation of LCMV-specific CD8^+^ T cells in the absence of IFNAR1.

A more comprehensive phenotypic analysis demonstrated that virus-specific CD8^+^ T cells in IFNAR1 KO and DKO mice, but not WT or PD-L1 KO mice, were functionally exhausted (Figure 2d,e). This was characterised by the enhanced presence of PD1, LAG3, and CD44 and the reduced presence of KLRG1 on virus-specific CD8^+^ T cells in IFNAR1 KO and DKO mice (Figure 2d,e). Further, after stimulation with LCMV-specific peptides, virus-specific CD8^+^ T cells in IFNAR1 KO and DKO mice showed a substantially reduced expression of IFN-γ, TNF, and granzyme B compared to WT and PD-L1 KO mice (Figure 3). Together, this implicates IFN-I and not PD-L1 signalling in promoting the anti-viral functions of CD8^+^ T cells in LCMV-Arm infection.

Similarly, virus-specific CD8^+^ T cells in IFNAR1 KO mice treated with and without anti-PD-L1 monoclonal antibody in the subacute (from day 15) or chronic (from day 24) phase of LCMV-Arm infection, also demonstrated a reduced expression of effector cytokines (Appendix A). Viral load in the liver and CNS, weight loss, and clinical score were additionally unaffected by anti-PD-L1 treatment (Appendix A). This suggests that the ablation or inhibition of PD-L1 has no effect on T cell reinvigoration in the absence of IFNAR1, highlighting the importance of IFN-I signalling to the accumulation of functional virus-specific CD8^+^ T cells in LCMV-Arm infection.

### 3.3. Ablation of IFNAR1 and PD-L1 Results in Enhanced Lung Pathology Following LCMV Infection

Both IFNAR1 KO and DKO mice exhibited comparable viral loads, with fewer virus-specific CD8^+^ T cells. Therefore, to explain the enhanced mortality and morbidity in DKO mice, we performed histology and PCR analysis on several tissues to examine differences in immunopathology and inflammatory cytokine expression (Figure 4).

Strikingly, relative to WT, PD-L1 KO, and IFNAR1 KO mice, DKO mice showed enhanced pathology in the lung (Figure 4). This was characterised by a substantial increase in the expression of interferon-stimulated genes (ISGs), PKR, ISG15, and IRF7 in the lung and ISG15 in the liver, but there were no changes in the CNS (Figure 4a). While these genes were also increased in the lungs of IFNAR1 KO, this was more pronounced in DKO mice (Figure 4a). The enhanced expression of these genes in IFNAR KO mice suggests they were induced by IFNAR-independent mechanisms [26,27,28,29,30]. Importantly, enhanced ISG expression in the lung of DKO mice translated to a massive inflammatory response, with the infiltration of polymorphonuclear cells and lymphocytes and oedema with an almost complete occlusion of the alveolar lumina in the lung (Figure 4b). In contrast, we observed only a mild increase in inflammatory foci in the liver of DKO mice, potentially in line with their increased ISG15 expression (Figure 4b).

To examine this response in more detail, we dissociated lungs from LCMV-infected DKO mice and characterised single-cell suspensions with high-dimensional spectral cytometry (Figure 5). The gating strategy used to identify lung subsets is shown in Appendix A. Confirming the presence of polymorphonuclear cells histologically (Figure 4b) by flow cytometry, we observed a substantial increase (9-fold) in the infiltration of neutrophils into the LCMV-infected DKO lung, compared to LCMV-infected WT lungs (Figure 5a,b). Conversely, we observed a significant reduction in the number of CD8^+^ T cells and a non-significant reduction in the number of eosinophils and NK cells in the lungs of DKO mice, compared to WT mice (Figure 5b). Interestingly, CD8^+^ T cells in LCMV-infected DKO lungs showed an enhanced expression of CD8α and PD-1 and reduced expression of Ly6C and CD69, compared to those from LCMV-infected WT mice (Figure 5c). This is illustrated by the shift of this population on the tSNE plot between genotypes (Figure 5a) and likely reflects a functionally exhausted phenotype. Together, the reduced infiltration of CD8^+^ T cells and the substantial increase in the number of neutrophils in DKO LCMV-infected lungs may explain the enhanced mortality observed in these mice (Figure 1a).

### 3.4. CD8^+^ T Cell Depletion but Not Neutrophil Depletion Rescues IFNAR1 x PD-L1 DKO Mice from Lethal LCMV Infection

To investigate the role of CD8^+^ T cells and neutrophils in the enhanced lung pathology and mortality seen in LCMV-infected DKO mice, we used mAbs to deplete these cells (Figure 6). To deplete neutrophils, we injected anti-GR1 in combination with anti-Ly6G. This depleted 93% of neutrophils in the lung and 92% in the blood (Figure 6a,b and Appendix A). Notably, while anti-GR1 also targets classic dendritic cells (cDCs), pDCs, and Ly6C^hi^ and Ly6C^lo^ monocyte-derived cells (MCs), reducing their infiltration into the lung (Figure 6b), these cells comprise a minor proportion of the DKO lung infiltrate relative to neutrophils (Figure 5b). Additionally, while using anti-Ly6G more specifically targets neutrophils, this was not effective at depleting these cells alone (Appendix A). Conversely, CD8^+^ T cells were efficiently depleted by 96% in the lung and 98% in the blood of DKO mice by injecting animals with an anti-CD8 mAb (Figure 6b and Appendix A).

Surprisingly, CD8^+^ T cell depletion rescued LCMV-infected DKO mice from lethal infection, while neutrophil depletion did not (Figure 6c). At day 9 p.i., anti-CD8 mAb-treated mice also showed a lower clinical score and reduced weight loss, while neutrophil depletion had no effect on these parameters (Figure 6d,e). This suggests that mortality and morbidity in DKO mice is largely driven by the infiltration of exhausted CD8^+^ T cells and not the massive 9-fold increase in the number of neutrophils in the lung, particularly since anti-CD8 mAb treatment rescued mice from lethal infection despite there being substantial neutrophil infiltrate in the lung (Figure 6a–c).

The examination of RNA levels demonstrated a decreased expression of granzyme B and an increased expression of TGF-β in the lung and liver of CD8^+^ T cell-depleted DKO mice, compared to neutrophil-depleted DKO mice (Appendix A). This may contribute to the reduced mortality in mice treated with anti-CD8. We also observed a significant reduction in IL-1β in the lungs of anti-GR1- and anti-Ly6G-treated DKO mice, suggesting that this cytokine is neutrophil-, MC-, or DC-derived (Appendix A). The expression of IL-1β, however, was not reduced in mice rescued from infection with anti-CD8 mAb, suggesting that this nominally pro-inflammatory cytokine does not contribute to lethal pathology. Nonetheless, to better understand the therapeutic effect of anti-CD8 treatment, we examined the histological changes in the lung. DKO mice show a massive infiltration of leukocytes and oedematous occlusion of the alveolar lumina in the lung (Figure 4b), similar to undepleted mice (Figure 6f). Intriguingly, CD8^+^ T cell depletion completely reversed this inflammatory response, likely explaining its protective effect in DKO mice (Figure 6f). The liver of anti-CD8-treated mice also showed a substantial reduction in lymphocyte infiltration. Taken together, these findings suggest that the infiltration of exhausted CD8^+^ T cells in the lungs of IFNAR1 and PD-L1 DKO mice, primarily induced by the lack of IFN-I signalling, is pathogenic. This pathogenic process promotes morbidity, mortality, and significant lung pathology.

## 4. Discussion

Murine models of LCMV infection using various virus strains and inoculation routes have been critical to our current understanding of type I interferons, anti-viral immunity, virus persistence, and functional T cell exhaustion. Accordingly, these discoveries have formed the foundation for the development of novel treatments for both acute and chronic viral infections and malignant tumours. Adding to this body of evidence as well as supporting previously published work, we have shown here that (1) IFNAR1 and not PD-L1 signalling is required to prevent CD8^+^ T cell exhaustion and virus persistence in LCMV-Arm infection, and (2) in the absence of functional IFN-I responses, PD-L1 protects from lethal CD8^+^ T cell-mediated lung inflammation.

Infection with LCMV-Arm results in an acute self-limiting infection, with the virus cleared within two weeks in immunocompetent mice [3,4,5,6,25]. By contrast, inoculation with LCMV-CL13 results in the exhaustion of CD8^+^ T cells, virus persistence, and chronic infection [4]. Intriguingly, the absence of functional IFN-I signalling in LCMV-Arm-infected mice also results in CD8^+^ T cell exhaustion and virus persistence [6], mirroring aspects of chronic infection with LCMV-CL13. In both LCMV-CL13- [4] and LCMV-Arm [5]-infected mice, CD8^+^ T cells upregulate PD-1 expression, and monoclonal antibody treatment targeting PD-L1, its ligand, during early infection invigorates the T response, promoting viral clearance [4,5]. In the current study, the absence of PD-L1 signalling did not reverse the exhausted CD8^+^ T cell phenotype observed in IFNAR1 KO mice. This suggests that IFNAR1 and not PD-L1 is required to generate a functional CD8^+^ T cell response in LCMV-Arm-infected mice.

The importance of IFNAR1 to a functional CD8 T^+^ cell response is well characterised. For instance, blocking IFNAR1 signalling with a monoclonal antibody in LCMV-CL13 infection results in a reduced frequency of virus-specific CD8^+^ T cells, which express PD-1 more highly [31]. Importantly, the administration of IFN-I can reverse this effect [32]. This exhausted CD8^+^ T cell phenotype has also been observed in murine gammaherpesvirus MHV68-infected IFNAR1 KO mice [33], highlighting the importance of IFNAR1 signalling to the differentiation of effector CD8^+^ T cells across different virus types. In addition to IFNAR1, we have also shown the importance of IRF9, a component of the IFN-I-induced signalling complex interferon-stimulated gene factor 3 (ISGF3), in functional T cell responses in LCMV-Arm-infection, with both IFNAR1 and IRF9 independently required in a CD8^+^ T cell-extrinsic manner [6]. Thus, the transfer of CD8^+^ T cells from a mouse with a transgenic T cell receptor (P14) recognizing the LCMV peptide GP33-41, into WT, but not IRF9 KO or IFNAR1 KO, LCMV-infected mice results in the accumulation of virus-specific CD8^+^ T cells with functional cytokine expression [6]. Further, in a separate study, WT CD8^+^ T cells that were adoptively transferred into WT LCMV-infected hosts expanded, while CD8^+^ T cells from IFNAR1 KO P14 did not [7]. This finding provides evidence that IFNAR1 is essential in a CD8^+^ T cell-intrinsic manner. Together, these studies strongly suggest that IFN-I signalling in CD8^+^ T cells as well as in other cells is critical for effector T cell responses in LCMV infection. This is because IFNAR1 promotes the clonal expansion and survival of CD8^+^ T cells, enabling the generation and accumulation of effector cells [6,7].

In the current study, we showed that in the absence of IFNAR1 signalling, PD-L1 was required for survival by preventing an overexuberant inflammatory response in the lung, while blockade of PD-L1 during later stages of infection had no adverse effect. This suggests a dual nature of PD-L1 in terms of its effect on disease outcome that is dependent on the stage of infection. This may not be unexpected given that PD-1 not only regulates anti-viral immunity but also protects the host against immunopathology [31,34]. Indeed, another study has also demonstrated lethal outcomes when the PD-1/PD-L1 pathway was ablated in LCMV-CL13 [4]. However, as previously mentioned, targeting this pathway can also be protective, reducing viral load and promoting T cell responses in LCMV-CL13 [4] and LCMV-Arm-infected [5] mice. Moreover, inhibiting PD-L1 in the early phase of infection (d0-8) results in lethal systemic tissue damage, while targeting PD-L1 in the latter part of the disease (d8-22) is protective [31]. This highlights the contrasting roles of PD-1-/PD-L1–mediated function in a time- and context-specific manner.

In our study, the depletion of CD8^+^ T cells reversed lethal lung pathology, indicating that CD8^+^ T cells mediated immunopathology directly or indirectly in DKO mice. Similarly, CD8^+^ T cell depletion prior to LCMV-CL13 infection rescued mice from PD-L1 blockade-mediated death [31]. This was also demonstrated in LCMV-docile-infected mice with a PD-L1/PD-1 signalling deficiency [35]. In this latter study, it was shown that PD-1 KO CD8^+^ T cells exhibited significantly enhanced functionality, compared with WT CD8^+^ T cells, also resulting in fatal lung pathology in LCMV-infected mice [35]. PD-L1 KO CD8^+^ T cells showed perforin-mediated killing of endothelial cells, ultimately leading to vascular permeability in the kidney, liver, lung, and brain; however, this was more pronounced in the lung, likely due to the large endothelial area in this organ [35]. Consistent with endothelial cell destruction, pathological changes were characterised by pulmonary oedema; thickened, oedematous septa; and scattered intra-alveolar fluid accumulation, together with a decrease in arterial blood pressure to levels insufficient for organ perfusion. Although further experiments are required, the findings demonstrated in Frebel et al., 2012, likely explain the enhanced lung pathology shown in DKO mice in our study. Together, this work highlights the critical immunoregulatory functions of the PD-1/PD-L1 pathway, with the ablation of this system resulting in systemic tissue damage and fatality.

In a model of intracerebral LCMV-Arm infection, CD8^+^ T cells mediated the infiltration of myelomonocytic cells, including neutrophils into the CNS, and contributed to seizure-induced death [36]. CD8^+^ T cells in DKO mice in our study may also have contributed to the infiltration of neutrophils in the lung, particularly since exhausted CD8^+^ T cells upregulate *Ccl3* and *Ccl4* [12], which have a role in neutrophil recruitment [37]. However, we did not observe a decrease in the number of neutrophils infiltrating the lung after CD8^+^ T cell depletion. Thus, the enhanced systemic levels of CCL5 more likely contributed to their recruitment [38]. Nonetheless, neutrophil depletion in DKO mice had no effect on survival or disease score, indicating that neutrophils are not required for disease development in these animals. This corresponds to previous findings showing that neutrophil depletion using anti-Ly6G or the ablation of INOS expression (highly expressed by neutrophils) did not alter the development of pathology in LCMV-infected PD-1 KO mice [35]. Moreover, anti-GR1 treatment in mice intracerebrally infected with LCMV-Arm did not improve survival [36]. Nevertheless, further work is required to determine whether neutrophil recruitment is a bystander effect of CD8^+^ T cell exhaustion and to what extent, once present, neutrophils may contribute to disease progression.

## 5. Conclusions

Overall, this work highlights the crucial interplay between PD-L1 and IFN-I signalling in LCMV-Arm infection for a controlled and functional CD8^+^ T cell response. Specifically, we demonstrate that IFNAR and not PD-L1 is required for the accumulation of virus-specific CD8^+^ T cells to prevent virus persistence and chronic LCMV-Arm infection, whereas in the absence of IFN-I signalling, PD-L1 is required to prevent an overexuberant CD8^+^ T cell response from causing severe lung damage and mortality. Intriguingly, the depletion of CD8^+^ T cells and not neutrophils rescued mice from lethal lung disease, demonstrating CD8^+^ T cell-mediated immunopathology in DKO mice. This has implications for therapeutics that target these pathways in the context of chronic infections and cancer.

## Figures and Tables

**Figure 1 viruses-16-00390-f001:**
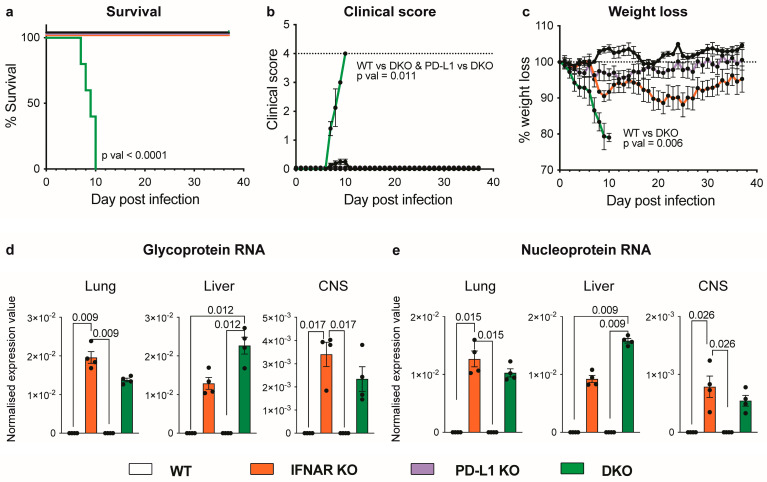
Absence of IFNAR1 and PD-L1 signalling results in lethal disease following LCMV infection. (**a**) Survival, (**b**) clinical score (rough fur = 1, hunched posture = 2, reduced activity = 2, tremor = 3, and seizures = 4) and (**c**) percent weight lost by wild-type (WT), IFNAR1 KO, PDLI KO and IFNAR1 x PD-L1 DKO mice infected intraperitoneally with LCMV-Arm (1 × 10^3^ PFU) and monitored daily for 38 days. Dotted lines indicate score requiring euthanasia (**b**) and 100% weight on day of infection (**c**). (**d**,**e**) RNA expression of LCMV-GP (**d**) and LCMV-NP (**e**) in the lung, liver and CNS of WT, IFNAR1 KO, PDLI KO and IFNAR1 x PD-L1 DKO mice at day 7 post-infection. Gene expression values were determined using qPCR and normalized to *18S*. Data are presented as mean ± SEM from one (**d**,**e**) or one of a minimum of two independent experiments (**a**–**c**) with at least four mice used per group and experiment. A Log-rank (Mantel–Cox) test was used to determine the statistical significance of survival data between all experimental groups (pval < 0.0001) and between two groups (WT vs. DKO, pval = 0.0065; PD-L1 KO vs. DKO, pval = 0.0065; IFNAR KO vs. DKO, pval = 0.0002). A Kruskal–Wallis test was used for clinical score (on day 9), weight loss (on day 9) and qPCR data (**b**–**d**).

**Figure 2 viruses-16-00390-f002:**
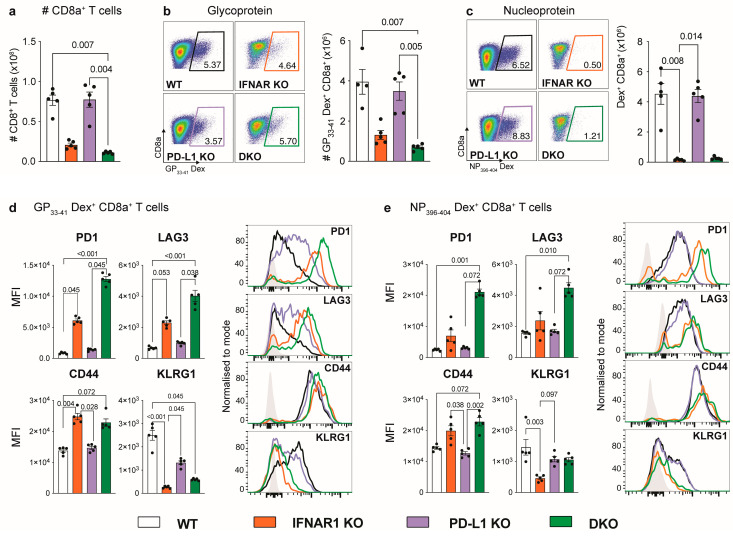
IFNAR1-deficient mice fail to accumulate LCMV-specific CD8^+^ T cells independent of PD-L1. (**a**) Number of CD8^+^ T cells, (**b**) LCMV GP33-41-Dex^+^CD8^+^ T cells and (**c**) LCMV NP396-404-Dex^+^CD8^+^ T cells isolated at day 8 p.i. from the spleens of WT, IFNAR1 KO, PDL1 KO and IFNAR1 x PD-L1 DKO mice, infected intraperitoneally with LCMV-Arm (2 × 10^5^ PFU). Representative dot plots of LCMV GP33-41-Dex^+^ CD8^+^ T cells (**b**) and NP396-404-Dex^+^ CD8^+^ T cells (**c**) are shown for WT (black gate), IFNAR1 KO (orange gate), PDL1 KO (purple gate) and IFNAR1 x PD-L1 DKO (green gate) spleens. (**d**,**e**) Bar plots and histograms showing the median fluorescence intensity for PD1, LAG3, CD44 and KLRG1 on LCMV GP33-41-Dex^+^CD8^+^ T cells (**d**) and LCMV NP396-404-Dex^+^ CD8^+^ T cells (**e**) in the spleen of LCMV-infected WT, IFNAR1 KO, PDL1 KO and IFNAR1 x PD-L1 DKO mice. Grey histogram shows the fluorescence minus one for each marker. Data are presented as mean ± SEM from one of two independent experiments with consistent results using at least four mice per group and experiment. A Kruskal–Wallis test was used to determine the statistical significance of flow cytometry data.

**Figure 3 viruses-16-00390-f003:**
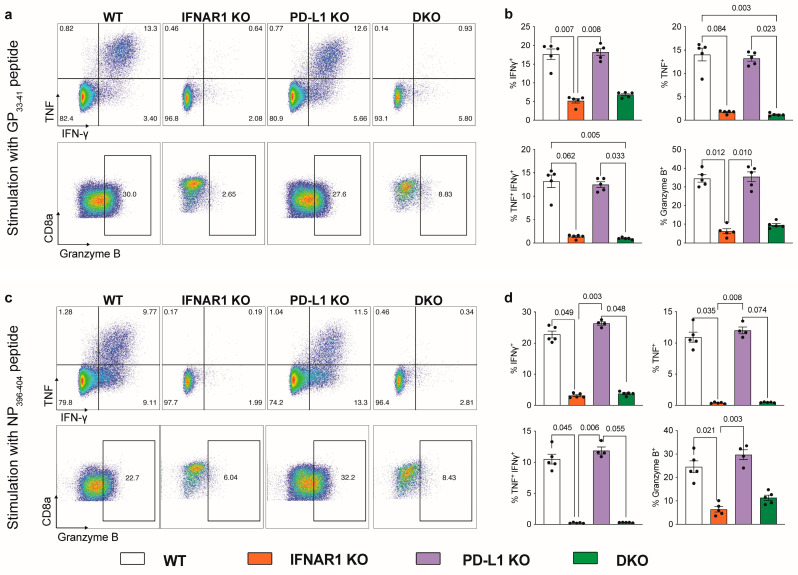
IFNAR1-deficient mice fail to accumulate functional CD8^+^ T cells expressing effector cytokines independent of PD-L1. (**a**–**d**) Dot plot (**a**,**c**) and a bar graph (**b**,**d**) showing the frequency of CD8^+^ T cells expressing IFN-γ, TNF and granzyme B after stimulation with LCMV GP33-41 (**a**,**b**) or NP396-404 (**c**,**d**). Splenocytes were isolated on day 8 p.i. with LCMV-Arm (2 × 10^5^ PFU). Data are presented as mean ± SEM from one of two independent experiments with consistent results using at least four mice per group and experiment. A Kruskal–Wallis test was used to determine the statistical significance of flow cytometry data.

**Figure 4 viruses-16-00390-f004:**
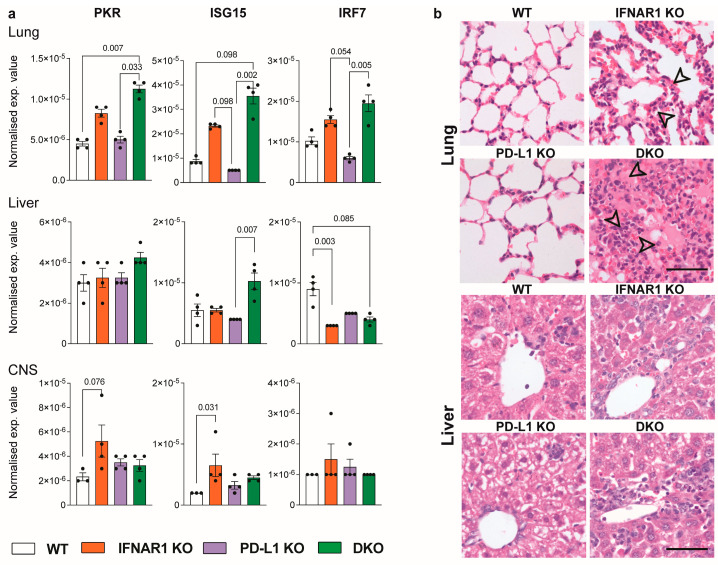
IFNAR1 x PD-L1 DKO mice show enhanced lung pathology following LCMV infection. (**a**) RNA expression of PKR, ISG15 and IRF7 in the lung, liver and CNS of WT, IFNAR1 KO, PD-L1 KO and IFNAR1 x PD-L1 DKO mice at day 8 post-LCMV infection (1 × 10^3^ PFU). Gene expression values were determined using qPCR and normalized to *18S.* (**b**) H&E images of the lung and liver from WT, IFNAR1 KO, PDL1 KO and IFNAR1 x PD-L1 DKO mice infected for 8 days with LCMV (1 × 10^3^ PFU). Arrows indicate inflammatory foci, including thickened septa between alveoli, proteinaceous oedema and infiltration of lymphocytic or polymorphonucleated cells. Scale bars = 50 μm. Data are presented as mean ± SEM from one of two independent experiments with consistent results using at least three mice per group, per experiment. A Kruskal–Wallis test was used to determine statistical significance.

**Figure 5 viruses-16-00390-f005:**
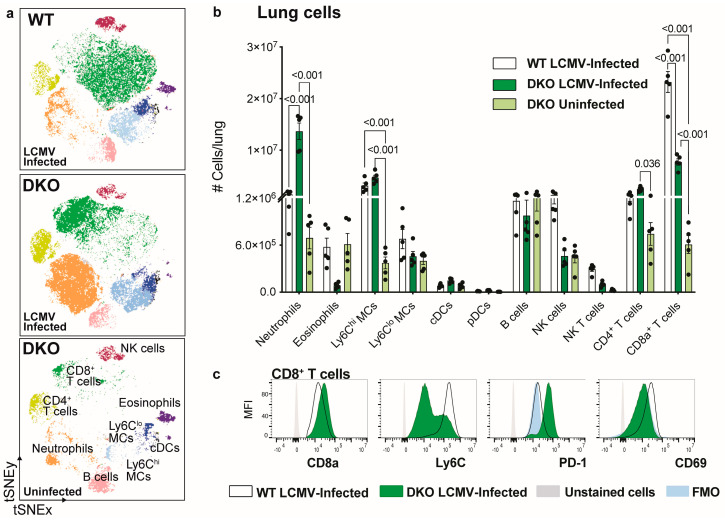
Enhanced lung pathology in IFNAR1 x PD-L1 DKO mice is associated with a 9-fold increase in the recruitment of neutrophils following LCMV infection. (**a**) tSNE plot clustered on live lung leukocytes (Live CD45^+^ cells) from LCMV-infected (2 × 10^3^ PFU; day 8 p.i.) WT, LCMV-infected DKO and uninfected DKO mice were culled at day 8 p.i. (**b**) Number of leukocyte subsets in the lung of LCMV-infected WT, LCMV-infected DKO and uninfected DKO mice culled at day 8 p.i. (**c**) Histograms showing the expression of select markers on CD8^+^ T cells in the lung of LCMV-infected WT and LCMV-infected DKO mice culled at day 8 p.i. Unstained cells are shown in grey and fluorescence minus one (FMO) is shown for PD-1 and CD69 in blue. Data are presented as mean ± SEM from one of two independent experiments with consistent results, using at least three mice per group per experiment. A Kruskal–Wallis test was used to determine statistical significance.

**Figure 6 viruses-16-00390-f006:**
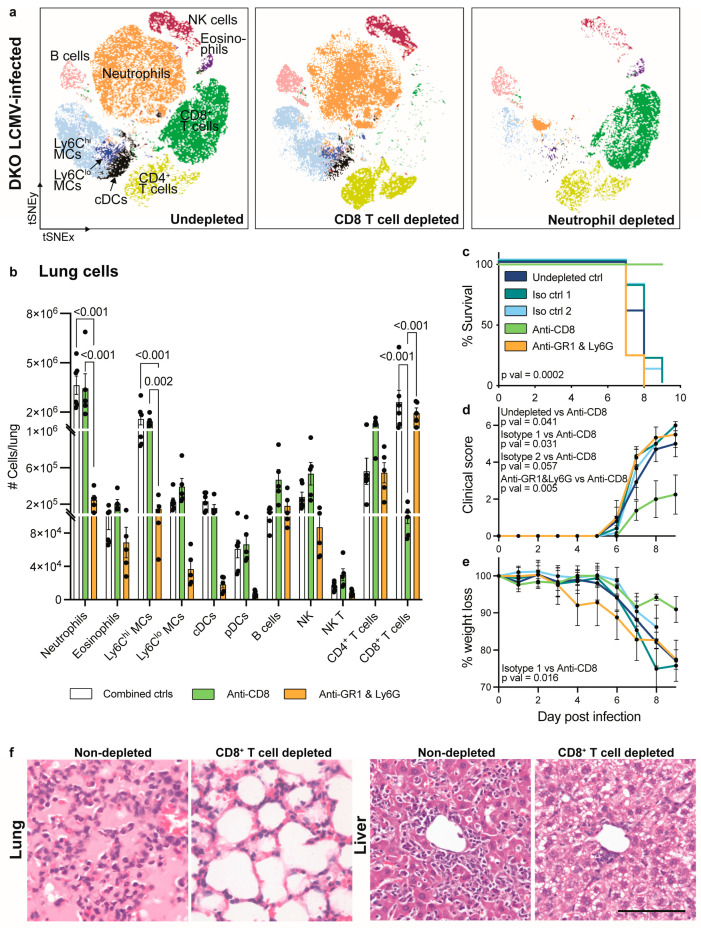
Depletion of CD8^+^ cells rescues LCMV-infected IFNAR1 x PD-L1 DKO mice. (**a**) tSNE plot clustered on live lung leukocytes (Live CD45^+^ cells) from LCMV-infected DKO mice untreated or treated with anti-CD8 or anti-GR1 and anti-Ly6G to deplete CD8^+^ T cells and neutrophils, respectively. Mice were infected with LCMV (2 × 10^3^ PFU) and euthanised on day 7 p.i. (**b**) Number of leukocyte subsets in the lung of non-depleted, CD8^+^ T cell-depleted and neutrophil-depleted mice. Undepleted mice and mice treated with one or two isotype controls were combined here, as there was no significant difference between these groups. (**c**) Survival, (**d**) clinical score (rough fur = 1, hunched posture = 2, reduced activity = 2, tremor = 3, and seizures = 4) and (**e**) percent weight lost by LCMV-infected IFNAR1 x PD-L1 DKO mice that were untreated, treated with one or two isotype control mAbs, an anti-CD8 mAB or anti-GR1 and anti-Ly6G mAbs. (**f**) H&E images of the lung and liver from LCMV-infected DKO mice treated with and without anti-CD8 mAb. Mice were infected with LCMV (1 × 10^3^ PFU) and euthanised on day 9 p.i. Scale bars = 50 μm. Data are presented as mean ± SEM from one independent experiment with 3–5 mice per group. A Log-rank (Mantel–Cox) test was used to determine the statistical significance of survival data (**c**) between all experimental groups (pval = 0.0002) and between two groups (undepleted vs. anti-CD8, pval = 0.0009; undepleted vs. anti-Ly6G, pval = 0.02; anti-Ly6G vs. anti-CD8, pval = 0.0014; isotype 1 vs. anti-CD8, pval = 0.0067; Isotype 2 vs. CD8, pval = 0.004). A Kruskal–Wallis test was used to determine the statistical significance of clinical score (on day 8), weight loss (on day 8) and flow cytometry data (**b**,**d**,**e**).

## Data Availability

The datasets used and/or analysed during the current study are available from the corresponding author upon reasonable request.

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
