# Peer review of "CD8+ T Cells Mediate Lethal Lung Pathology in the Absence of PD-L1 and Type I Interferon Signalling following LCMV Infection"

_viruses, 2024, doi:10.3390/v16030390_

Round 1

Reviewer 1 Report

Comments and Suggestions for Authors

Overall, this study is well-designed and their findings also significant. Thus, I only have two minor comment.

1. the present abstract lacks clarity in stating the aim and method of the study. I recommend revising the first part of the abstract to address this.

2. In the last paragraph, the authors provide a brief summary of their findings. It is advisable to conclude the introduction with a restatement of the study's aim and remove the specific findings

Author Response

Overall, this study is well-designed and their findings also significant. Thus, I only have two minor comments.

We thank the reviewer for their positive comments.

  1. The present abstract lacks clarity in stating the aim and method of the study. I recommend revising the first part of the abstract to address this.

We agree with this comment and accordingly have rewritten our abstract to make it clearer and to better outline the aim of this study.

Abstract (lines 17-31): " CD8+ T-cells are critical to the adaptive immune response against viral pathogens. However, overwhelming antigen exposure can result in their exhaustion characterised by reduced effector function, failure to clear virus and the upregulation of inhibitory receptors, including programmed cell death 1 (PD-1). However, exhausted T-cell responses can be “re-invigorated” by inhibiting PD-1 or the primary ligand of PD-1, PD-L1. Further, absence of the type I interferon receptor IFNAR1 also results in T-cell exhaustion and virus persistence in lymphocytic choriomeningitis virus Armstrong (LCMV-Arm) infected mice. In this study, utilizing single and double knock out mice, we aimed to determine whether ablation of PD-1 could restore T cell functionality in the absence of IFNAR1 signalling in LCMV-Arm infected mice. Surprisingly, this did not re-invigorate the T-cell response and instead, it converted chronic LCMV-Arm infection into a lethal disease characterized by severe lung inflammation with infiltration of neutrophils and T-cells. Depletion of CD8+ T-cells, but not neutrophils, rescued mice from lethal disease, demonstrating that IFNAR1 is required to prevent T-cell exhaustion and virus persistence in LCMV-Arm infection, and in the absence of IFNAR1, PD-L1 is required for survival. This reveals an important interplay between IFNAR1 and PD-L1 with implications for therapeutics targeting these pathways."

  1. In the last paragraph, the authors provide a brief summary of their findings. It is advisable to conclude the introduction with a restatement of the study's aim and remove the specific findings.

We thank the reviewer for this comment and have removed the specific findings from the introduction, as well as restated the aims of the study. 

Introduction (lines 75-83): "As outlined above, we have previously shown that IFNAR1 signaling is critical to the generation of functional CD8+ T cells in LCMV-Arm infection. Therefore, in the current study we aimed to investigate the role of PD-L1 signaling in the generation of exhausted T cells and virus persistence in LCMV-infected IFNAR1 knockout (KO) mice. Utilizing IFNAR KO, PD-L1 KO and double KO animals (PD-L1 and IFNAR KO, DKO), we reveal that IFNAR1 and not PD-L1 is required for a functional CD8+ T cell response, while PD-L1 was required to prevent lethal lung pathology in IFNAR1 KO mice. This study reveals a significant interplay between IFNAR1 and PD-L1 signaling, highlighting its implications for therapeutics that target these pathways in the context of chronic infections and cancer."

Reviewer 2 Report

Comments and Suggestions for Authors

This manuscript submitted by Spiteri et al., described that the lung pathology could be observed upon the LCMV infection in PD-L1 and type I interferon signaling double knocked out mice, and this is due to the CD8+ T cells.

Lymphocytic Choriomeningitis virus (LCMV) is a powerful tool to understand the acute and persistent viral infection. Rapid virus clearance is observed by the Armstrong strain of LCMV infection in C57BL/6 mice, while clone 13 strain of LCMV infection in C57BL/6 mice causes persistent infection. This difference could be explained by the activation of cytotoxic T lymphocyte (CTL or CD8(+) T cells). In case of the Armstrong strain infection, CTL undergoes the clonal expansion and activation to clear the virus, while in case of the clone 13 infection, CTL exhibited exhaustion which fails to clear the virus. PD-1 is one of the critical inhibitory molecules in the CD8(+) T cells for the exhaustion, especially in case of the LCMV infection. In addition to this inhibitory molecules, type I interferon signaling was also shown to be important to activate the CTL, followed by the clearance of the virus.

In this study, authors showed that lack of the type I interferon signaling and PD-L1 induced lethal lung pathology upon the LCMV Armstrong strain infection, and this is due to the CD8(+) T cells. Although the single lack of type I interferon signaling, but not the PD-L1, increased viral load in lung, liver, and in CNS, this did not affect to the mice lethality and pathology. When the same infection was conducted with IFNAR1/PD-L1 double KO mice, it showed 100% mortality with the severe lung pathology. CD8(+) T cells were shown to be the main cause of the lung pathology infected by the virus in IFNAR1/PD-L1 double KO mice. The series of the experiments clearly showed that PD-1-PD-L1 interaction is important for suppressing excess CD8(+) T cell attack to the lung.

The concept is clear and the overall experiments were conducted properly with appropriate description. However, reviewer requested to add some discussion and information to improve the manuscript as below.

1.     All infection experiments were conducted with the i.p. injection. The main procedure for the LCMV infection, especially focusing on acute vs persistent, has been conducted with 2 x 10^6 i.v. infection. Please include the explanation why the infection was conducted by i.p. not by i.v., and what kind of difference would be expected.

2.     Upon the i.p. infection, the authors found a clear pathological effect in the lung. Please include the explanation why they could not observe similar pathological effect in other organs. Also, please include the explanation of the relevance on focusing lung pathology upon the i.p. infection of LCMV. Are there any examples which show lung pathology upon the infection not by the route of respiratory tract such as i.n., but by the other routes such as i.v., i.p. etc.?

3.     In Fig.4a, authors observed upregulation of ISGs in IFNAR1 KO mice upon the infection. Please include the discussion how this could be happened? Is there any known signaling pathway which induces ISGs’ expression except for the canonical type I interferon signaling pathway?

Minor points

4.     In Figure 2., the description of WT, IFNAR1 KO, PD-L1 KO, and DKO is not shown as does in Figure 3.

5.     Reviewer wonder if the description of X-axis and Y-axis in Figure S3 (left bottom) is opposite (CD4 for Y-axis and CD8 for X-axis?).

Author Response

The concept is clear and the overall experiments were conducted properly with appropriate description. However, reviewer requested to add some discussion and information to improve the manuscript as below.

 We thank the reviewer for taking the time to review our manuscript and providing these positive comments.

  1. All infection experiments were conducted with the i.p. injection. The main procedure for the LCMV infection, especially focusing on acute vs persistent, has been conducted with 2 x 10^6 i.v. infection. Please include the explanation why the infection was conducted by i.p. not by i.v., and what kind of difference would be expected.

In mouse models of LCMV infection, the chosen inoculation route depends on the strain of virus. Typically, clone-13 infections are done via an intravenous route to achieve chronic infection, while Armstrong (used in this study) is injected both intravenously and intraperitoneally 1-3. Here, an intraperitoneal injection was used to compare findings more directly to our previous studies investigating IFN-I signalling 4.

Material and Methods (lines 102-103): "This route of infection was used to allow for better comparison to our previous study 4."

  1. Upon the i.p. infection, the authors found a clear pathological effect in the lung. Please include the explanation why they could not observe similar pathological effect in other organs. Also, please include the explanation of the relevance on focusing lung pathology upon the i.p. infection of LCMV. Are there any examples which show lung pathology upon the infection not by the route of respiratory tract such as i.n., but by the other routes such as i.v., i.p. etc.

We thank the reviewer for this helpful comment and provided an explanation for the changes seen in the lung. Importantly, Frebel et al., 2012 demonstrated lethal lung pathology following i.v. infection of LCMV in PD-1 knockout mice. This strongly suggests that the pathology observed in this study is independent of inoculation route. Intriguingly, Frebel et al., revealed extensive vascular permeability and alterations in multiple organs with changes being most pronounced in the lung suggesting that the lung is in particular vulnerable to LCMV. They further showed that endothelial cells were killed by CD8+ T cells in the absence of PD-1 signalling, promoting vascular permeability, oedema and association pathology in the lung. It was thought that the effect was more prominent in the lung because it has the largest endothelial area or because lung endothelium is more affected than other endothelia. We have used these results to further explain the pathological effects seen in the lung in our study.

Discussion (lines 480-490): "In this latter study, it was shown that PD-1 KO CD8+ T cells exhibited significantly enhanced functionality, compared with WT CD8+ T cells, also resulting in fatal lung pathology in LCMV-infected mice 5. PD-L1 KO CD8+ T cells showed perforin-mediated killing of endothelial cells, ultimately leading to vascular permeability in the kidney, liver, lung and brain, however, this was more pronounced in the lung, likely due to the large endothelial area in this organ 5. Consistent with endothelial cell destruction, pathological changes were characterised by pulmonary oedema, thickened, oedematous septa and scattered intra-alveolar fluid accumulation, together with a decrease in arterial blood pressure to levels insufficient for organ perfusion. Although further experiments are required, the findings demonstrated in Frebel et al., 2012, likely explain the enhanced lung pathology shown in DKO mice in our study."

  1. In Fig.4a, authors observed upregulation of ISGs in IFNAR1 KO mice upon the infection. Please include the discussion how this could be happened? Is there any known signaling pathway which induces ISGs’ expression except for the canonical type I interferon signaling pathway?

Induction of interferon-stimulated genes is not necessarily specific for IFN-I signalling 6-9, with ISG15 also induced by infection, lipopolysaccharide, retinoic acid and genotoxic stressors 10. We have now included this explanation in the results section, please see lines 314-316.

Results (lines 316-318): "The enhanced expression of these genes in IFNAR KO mice suggests these were induced by IFNAR-independent mechanisms 6-9. Indeed, ISG15 is also induced by infection, lipopolysaccharide, retinoic acid and genotoxic stressors 10."

Minor points

  1. In Figure 2., the description of WT, IFNAR1 KO, PD-L1 KO, and DKO is not shown as does in Figure 3.

Thank you for pointing this out. The figure has been amended.

  1. Reviewer wonder if the description of X-axis and Y-axis in Figure S3 (left bottom) is opposite (CD4 for Y-axis and CD8 for X-axis?).

Response: We thank the reviewer for pointing this mistake out and have now corrected this.

References

  1. Suprunenko T, Hofer MJ. Complexities of Type I Interferon Biology: Lessons from LCMV. Viruses 2019; 11.
  2. Bocharov G, Argilaguet J, Meyerhans A. Understanding Experimental LCMV Infection of Mice: The Role of Mathematical Models. J Immunol Res 2015; 2015: 739706.
  3. Dangi T, Chung YR, Palacio N, Penaloza-MacMaster P. Interrogating Adaptive Immunity Using LCMV. Curr Protoc Immunol 2020; 130: e99.
  4. Huber M, Suprunenko T, Ashhurst T, et al. IRF9 Prevents CD8(+) T Cell Exhaustion in an Extrinsic Manner during Acute Lymphocytic Choriomeningitis Virus Infection. J Virol 2017; 91.
  5. Frebel H, Nindl V, Schuepbach RA, et al. Programmed death 1 protects from fatal circulatory failure during systemic virus infection of mice. J Exp Med 2012; 209: 2485-2499.
  6. Ashley CL, Abendroth A, McSharry BP, Slobedman B. Interferon-Independent Innate Responses to Cytomegalovirus. Front Immunol 2019; 10: 2751.
  7. Ashley CL, Abendroth A, McSharry BP, Slobedman B. Interferon-Independent Upregulation of Interferon-Stimulated Genes during Human Cytomegalovirus Infection is Dependent on IRF3 Expression. Viruses 2019; 11.
  8. Cho SD, Shin H, Kim S, Kim HJ. Insights on interferon-independent induction of interferon-stimulated genes shaping the lung's response in early SARS-CoV-2 infection. Heliyon 2023; 9: e22997.
  9. Platanias LC. Mechanisms of type-I- and type-II-interferon-mediated signalling. Nat Rev Immunol 2005; 5: 375-386.
  10. Perng YC, Lenschow DJ. ISG15 in antiviral immunity and beyond. Nat Rev Microbiol 2018; 16: 423-439.
